# Language Statistics at Different Spatial, Temporal, and Grammatical Scales

**DOI:** 10.3390/e26090734

**Published:** 2024-08-29

**Authors:** Fernanda Sánchez-Puig, Rogelio Lozano-Aranda, Dante Pérez-Méndez, Ewan Colman, Alfredo J. Morales-Guzmán, Pedro Juan Rivera Torres, Carlos Pineda, Carlos Gershenson

**Affiliations:** 1Facultad de Ciencias, Universidad Nacional Autónoma de México, Mexico City 04510, Mexico; fernanda@ciencias.unam.mx (F.S.-P.); rogelio97@ciencias.unam.mx (R.L.-A.); 2Centro de Ciencias de la Complejidad, Universidad Nacional Autónoma de México, Mexico City 04510, Mexico; dante.perez.m@gmail.com (D.P.-M.); ecolman@exseed.ed.ac.uk (E.C.); pjr62@cam.ac.uk (P.J.R.T.); 3Instituto de Fisica Interdisciplinar y Sistemas Complejos, Universidad de las Islas Baleares, 07122 Palma de Mallorca, Spain; 4Posgrado en Ciencias de la Computación, Universidad Nacional Autónoma de México, Mexico City 04510, Mexico; 5Roslin Institute, University of Edinburgh, Midlothian EH8 9YL, UK; 6MIT Media Lab, Cambridge, MA 02139, USA; alfredo.mo.gu@gmail.com; 7Instituto de Física, Universidad Nacional Autónoma de México, Mexico City 04510, Mexico; carlospgmat03@gmail.com; 8School of Systems Science and Industrial Engineering, Binghamton University, Binghamton, NY 13902, USA; 9Instituto de Investigaciones en Matemáticas Aplicadas y Sistemas, Universidad Nacional Autónoma de México, Mexico City 04510, Mexico; 10Santa Fe Institute, Santa Fe, NM 87501, USA

**Keywords:** Twitter, rank diversity, geolocalization, ngrams, language models, statistics, scaling laws, complexity

## Abstract

In recent decades, the field of statistical linguistics has made significant strides, which have been fueled by the availability of data. Leveraging Twitter data, this paper explores the English and Spanish languages, investigating their rank diversity across different scales: temporal intervals (ranging from 3 to 96 h), spatial radii (spanning 3 km to over 3000 km), and grammatical word ngrams (ranging from 1-grams to 5-grams). The analysis focuses on word ngrams, examining a time period of 1 year (2014) and eight different countries. Our findings highlight the relevance of all three scales with the most substantial changes observed at the grammatical level. Specifically, at the monogram level, rank diversity curves exhibit remarkable similarity across languages, countries, and temporal or spatial scales. However, as the grammatical scale expands, variations in rank diversity become more pronounced and influenced by temporal, spatial, linguistic, and national factors. Additionally, we investigate the statistical characteristics of Twitter-specific tokens, including emojis, hashtags, and user mentions, revealing a sigmoid pattern in their rank diversity function. These insights contribute to quantifying universal language statistics while also identifying potential sources of variation.

## 1. Introduction

Statistical linguistics has emerged as a significant area of study in the past century [1], with Zipf’s law standing as a cornerstone in understanding the statistical distribution of rank–frequency distributions within languages. This law explains a fundamental relationship between the rank order and frequency of occurrence of words, asserting that the frequency of a word is inversely proportional to its rank when ranked by frequency.

In this context, the comparison between random text models and real texts has been pivotal, revealing that genuine texts comprehensively span the entire lexical spectrum irrespective of word length, thus underscoring the profound meaningfulness embedded within Zipf’s law [2,3,4,5,6]. Concurrently, random text models have been posited as explanatory constructs for Zipf’s law [7].

Several studies have been possible only recently because of the availability of large amounts of data (see the next Section). When enough data are available, not only statistical studies are possible for a single moment but also the dynamics of language, i.e., how language statistics change in time, are feasible as well. In this work, we investigate language usage dynamics across temporal, spatial, and grammatical scales using geolocated Twitter data.

Our aim is to determine the significance of each scale in language statistics, evaluating how changes in these scales influence rank dynamics.

Analyzing language change over time intervals enhances our understanding of language variations and its influencing factors. This method is valuable for examining geographical, grammatical, and temporal language variants, revealing how languages differ across different regions and periods.

Our research focuses on the differences of language usage at different time scales. By examining these temporal variations, we aim to uncover factors that drive linguistic change, contributing to a deeper understanding of language dynamics.

Smaller regions may exhibit less word diversity due to limited statistical sampling, and our analysis seeks to observe these differences across various scales. Despite a common language, conversation topics vary daily and regionally, and our statistical analyses effectively characterize these differences.

The work is mostly concerned with understanding what factors contribute to rank diversity, and the conclusions are mostly about this.

To our knowledge, this is the first study of its kind. Discovering the answers to our questions should be interesting, even if the results are not surprising. We believe that the importance of rank diversity (and other measures) will only become clear after multiple studies are conducted to evaluate its usefulness. However, we cannot assume its usefulness first and then interpret the data accordingly.

## 2. Related Work

Some studies have delved into the origins and evolution of language [8,9,10,11], leveraging the accessibility of vast datasets such as digitized books and the proliferation of social media. These datasets facilitate an in-depth exploration of language dynamics and evolutionary patterns over time [12,13,14].

Almodaresi et al. [15] have characterized various lexical distributions across different analytical levels, demonstrating their adherence to a log-normal distribution at both user and country levels.

Prior research has scrutinized language variations across time scales spanning years and centuries [12,16], offering valuable insights into diverse fields including lexicography, grammar evolution, collective memory, technology adoption, censorship, historical epidemiology, and the lifespan of commonly used words and phrases.

Other authors have conducted statistical studies showing the differences between Spanish and English. In [17], researchers analyzed large text samples from both languages and tested various methods to identify the language of a short text sample. The Kolmogorov–Smirnov Goodness-of-Fit test was found to be the most reliable, with a significance level of 0.0077 for a sample of around 107 characters (about 21 words). This research, as well as ours, supports the idea that each language has distinct statistical characteristics that enable identification from a short text sample.

Twitter data have been extensively utilized for analyzing language in various contexts such as sentiment analysis, topic analysis, (mis)information dissemination, and activity patterns [14,18,19,20,21,22]. Additionally, it has been instrumental in understanding global synchronization patterns [21], spreading mechanisms [20], and political polarization [23]. Leveraging metadata encompassing location, time, and text, we can analyze dynamics and geography across multiple scales. Previous research has identified variations in text usage and interaction mechanisms across different countries or cultures, including differences in the usage of URLs, hashtags, mentions, replies, and retweets [24,25]. Moreover, studies have shown that Twitter hashtags exhibit patterns consistent with the geographical landscape, reflecting the virtual space’s representation of the physical world [26]. Similar investigations have been conducted on characterizing rank dynamics in the Chinese microblog Sina Weibo [27], focusing on metrics such as the time spent by hashtags on the list, the timing of their appearance, rank diversity, and ranking trajectories.

Our previous studies [26,28] have revealed a consistent pattern of changes in word usage across various languages, which we quantified using a metric called *rank diversity*. This metric, calculated by examining a corpus of words ranked by frequency within specific time intervals, captures the variability in the occupancy of word ranks over time. When a rank consistently features the same word, rank diversity is low, whereas if different words occupy the same rank across time intervals, rank diversity is high. By plotting rank diversity against rank, we can analyze temporal changes in word usage. Our previous research [28,29,30] has shown that the relationship between rank diversity and rank can be approximated by a sigmoid function with similar parameters observed across different languages.

Twitter data present unique characteristics that make it a compelling subject of study. Unlike books or longer written texts, tweets are limited to a small number of characters, making them an intriguing medium for exploring language usage and its statistical differences from longer texts. Additionally, Twitter provides a finer temporal resolution compared to physically published materials, and its geotagging feature allows for the examination of geographical variations in language use at very fine scales. Moreover, the social network nature of Twitter, including user interactions such as mentions and retweets, as well as trending topics, creates a distinct language ecosystem. Finally, Twitter offers a vast dataset suitable for conducting comprehensive statistical analyses.

In this study, we conducted an analysis of over 20 million geolocated tweets originating from eight different countries. Our investigation focused on examining rank diversity across a range of spatial, temporal, and grammatical scales. The primary aim was to evaluate how variations in these scales impact rank diversity and to elucidate their implications for language dynamics.

While previous research has highlighted the statistical similarities between punctuation marks and words, as discussed in [31] and related literature, our study took a different approach. Since Twitter users, especially from the time when there was a 144-character limit, tended to disregard punctuation marks. Still, this should not make a difference in our analysis, as we are not analyzing rank distributions (that could be described with Zipf’s law) but rather rank dynamics. Thus, we concentrated solely on analyzing linguistic patterns involving Ngrams. This decision allowed us to delve deeper into the nuances of language dynamics across different scales without the confounding influence of punctuation. Moreover, this allowed us to compare directly results with a previous analysis of the Google Ngrams dataset, which also lacks punctuation marks.

Statistical language modeling has been used for information retrieval by treating each document as a language sample and a query as a generation process [32]. This way, language models can be used as probability distributions where documents are ranked based on the probability of generating the query from their language models.

## 3. Methods

### 3.1. Rank Diversity

We define rank diversity d(k) as the number of words occupying a given rank *k* during the period of time of the study, which is divided by the number *T* of time intervals considered.

Therefore, rank diversity is given by
(1)$$d\left(k\right)=\frac{\left|X(k)\right|}{T},$$
where |X(k)| is the cardinality (i.e., number of unique words) that appear at rank *k* during all time intervals.

Rank dynamics have been studied fields of sports [33] and other systems like natural, social, economic, and infrastructural systems. The authors of [34] precisely measured sampling effects on different measures of rank dynamics.

If a rank is occupied by a single word at all times, then rank diversity is minimal. While if in each time interval we have a different word in a given rank, then the rank diversity is maximum.

Rank diversity is an intuitive and straightforward measure of diversity within a dataset, providing direct insights into the variety and spread of elements. It is computationally efficient and robust, making it suitable for large datasets. In contrast, the Kullback–Leibler (KL) divergence is excellent for comparing probability distributions but can be less intuitive when choosing the base of the logarithm and the units of measurement, and it is sensitive to outliers. Rank correlation coefficients measure the similarity between two rankings but do not address diversity within a single dataset. Therefore, while KL divergence and rank correlation coefficients have their strengths, rank diversity offers unique benefits, especially in contexts focused on understanding the richness and variability within single datasets, such as linguistic studies.

### 3.2. Data

We used geolocated Twitter data to examine shifts in language usage patterns. These tweets were gathered using the Twitter Streaming Application Programming Interface (API), which serves as an intermediary mechanism facilitating data transfers between systems. Among the vast volume of collected tweets, only a subset is geolocalized. Our dataset comprises over 20 million geolocated tweets originating from Argentina, Canada, Colombia, India, Mexico, South Africa, Spain, and the United Kingdom, which are all posted in 2014. Notably, this was during a period when tweets were restricted to 140 characters, and threaded conversations were not as prevalent. We computed rank diversity metrics using various time intervals (Δt). It is worth mentioning that mentions and hashtags were included in the dataset and treated as separate words. These geolocated tweets contain precise latitude and longitude coordinates indicating their geographical origin at the time of creation.

Ferrer and Cancho showed in [35] that the word frequency spectrum typically follows a power law with an exponent around β=2, although this can vary significantly. This variation reflects how we balance effective communication (maximizing information transfer). The more we prioritize effective communication, the higher the exponent. The range of exponents goes from about β=1.6 (lower cost) to β=2.4 (higher cost). We performed the analysis of Zipf’s exponents in Figure 1.

### 3.3. Grammatical Scale

The grammatical scale, as defined in our study, pertains to the length of *n*-gram blocks utilized, where *n* ranges from 1 to 5 [16]. Single words are considered monograms or 1-grams, while sets of five words are termed 5-grams, and so forth.

The use of sequences of 5-grams has a compatibility reason: since the Google *n*-grams dataset includes up to 5-grams, our current study enables comparisons with previous research (ours and from others) that have used this dataset.

These *n*-gram blocks are constructed by partitioning sentences into words and assembling sets of contiguous words of length *n* with overlaps. For instance, the sentence “This is a sentence” yields three bigrams: (“This is”, “is a”, and “a sentence”).

In our previous research, we investigated how the grammatical scale impacts the rank dynamics of words using the Google Books *n*-gram dataset [29]. Our findings revealed that changes in the grammatical scale have a more pronounced effect on language statistics (such as rank diversity, change probability, rank entropy, and rank complexity) compared to alterations in the language itself (across English, Spanish, French, German, Italian, and Russian).

### 3.4. Temporal Scale

To delineate the temporal scale, it is essential to recall that we define rank diversity d(k) as the count of words occupying a particular rank *k* across all time intervals, which is divided by the total number of time intervals *T* considered. By altering the time interval Δt, we can compute the rank diversity for various values of *T*. It is worth noting that if the same dataset is utilized, as the temporal scale Δt increases, *T* will decrease.

The temporal scale undergoes variation by doubling its value at each time interval, i.e., 3,6,9,12,24,48,96 h. Evans and Larsen-Freeman [36] conducted a study on language acquisition across different temporal scales, yielding noteworthy findings.

To elucidate the differences of word ranks (specifically 1-grams, although the principle applies to any grammatical scale), we provide examples of selected Spanish words in Figure 2. For instance, in this scenario, d(k=1) is calculated by dividing the count of unique series (representing words graphically) that intersect the line denoting point k=1 at some time by the total number of 3 h intervals within a year.

In Figure 3, we present the sigmoid curves for 1, 2, 3, 4, and 5-grams for a 3 h time interval, which is the minimum time unit. We used a statistically significant sample size to ensure comparability, making sure that the smallest set is sufficiently large. Shorter time (and distance) intervals would produce too few data with the Twitter dataset we worked with.

### 3.5. Spatial Scale

We investigate the impact of the spatial scale using tweets exclusively from Mexico, Spain, Argentina, and the United Kingdom, as these countries possess sufficient geolocalized data to achieve statistical significance across various spatial scales (approximately 3.9, 3.7, 4.6, and 5.6 million tweets from Mexico, Spain, Argentina, and the United Kingdom, respectively).

Initially, we established a 3 km radius circle centered on the capital city’s geographical midpoint (Mexico City, Madrid, Buenos Aires, and London). Subsequently, we expanded the circle’s radius exponentially by powers of two, i.e., 6 km, 12 km, 24 km, 48 km, 98 km, and so forth, until encompassing the entire country.

Our decision to designate the distance from an urban hub (the capital city of each country) rather than the country’s geographic center aims to ensure a minimum population density, guaranteeing a sufficient number of tweets within a confined area. Initiating from the geographic center might yield areas lacking adequate data for statistical significance or comparability across countries. Additionally, countries were demarcated using polygons to exclude data from neighboring countries even if they fell within the considered radii.

To mitigate potential biases, we maintain an identical number of tweets across all spatial scales subsequent to the analysis of the smallest scale for each country. For instance, the quantity of tweets within the 3 km radius circle for Mexico totaled 309,792. Consequently, for all subsequent spatial scales, denoted by expanding-radius circles, we randomly sample 309,792 tweets without replacement to ensure uniform analysis conditions.

We conducted sampling to ensure it is statistically significant and comparable, so the smallest set is sufficiently large. Figure 4 illustrates how geographic distribution was modeled as a expanding circle considering the capital city as center of the circle and using Spain as an example.

### 3.6. Relevance of Scales

Since we are examining three distinct scales, namely grammatical, temporal, and spatial, to discern any unclear differences in the behavior of rank diversity across these scales, we generated a total of I·J·S rank diversity curves for each country. Here, *I*, *J*, and *S* denote the number of different values that a particular scale can adopt, representing grammatical, temporal, and spatial scales, respectively. Consequently, we generated rank diversity curves corresponding to every unique combination of values from the considered scales. For instance, in the case of Mexico, this equates to 5·6·11=330 possible combinations.

To obtain a quantitative summary of a rank diversity curve’s behavior, which measures how rapidly rank diversity increases relative to rank, and thereby simplify the system’s description and reduce observed complexity, we employed estimates of μ. μ represents a parameter of the sigmoid curve indicating the rank value at which the rank diversity curves reach 1/2.

The sigmoid is the cumulative of a Gaussian distribution, i.e.,
(2)$$\Phi_{\mu ,\sigma }\left(\log_{10}k\right)=\frac{1}{\sigma \sqrt{2\pi }}\int_{-\infty }^{\log_{10}k}e^{-\frac{{\left(y-\mu \right)}^{2}}{2\sigma^{2}}}dy,$$
and is given as a function of logk [28].

The fundamental concept for gauging the significance of scale changes on rank diversity behavior lies in recognizing that a lower value of μ signifies a swifter increase in rank diversity concerning rank. It is crucial to note that rank diversity, being a statistical metric, undergoes further averaging with μ, necessitating caution in result interpretation.

To assess the relative importance of a scale concerning the extent to which a transition between two distinct values of that scale impacts variations in rank diversity behavior, we used the following average:(3)$$\eta \left(s\right)=\frac{\sum_{i}^{I}\sum_{j}^{J}\sigma_{i,j}^{s}}{I\cdot J},$$
where $\sigma_{i,j}^{s}$ corresponds to the standard deviation of estimated values of μ associated to the scale *s* given fixed *i* and *j* values of the two remaining scales, i.e., if
(4)$${\bar{\mu }}_{i,j}=\sum_{s}\frac{\mu_{i,j}^{s}}{N},$$
then
(5)$$\sigma_{i,j}^{s}=\sqrt{\frac{\sum_{s}{\left(\mu_{i,j}^{s}-{\bar{\mu }}_{i,j}\right)}^{2}}{N-1}}.$$

In essence, η(s) aims to encapsulate the average dispersion of μ within the scale *s*, enabling an objective comparison of different scales to determine which one wields the greatest influence on modifying the rate of rank diversity increase across different scale values. In the Results section, we illustrate how η(s) effectively quantifies visually discernible trends using graphs of μ plotted against scale values.

This way, high η values indicates a strong association between the scales, meaning that the group means differ significantly. Meanwhile, low η values indicates a weak association, meaning the group means do not differ significantly.

Furthermore, to substantiate the visually observed trends with statistical evidence, we employed a linear regression model to conduct the *F*-test, evaluating whether at least one of the scales significantly impacts μ and consequently rank diversity. Additionally, the *t*-test of each coefficient associated with independent variables was utilized to ascertain whether individual scales contribute significantly to explaining the variability of μ within a linear model. In cases involving coefficients representing multiplicative terms in a multiplicative model, the *t*-test was employed to determine the presence of statistically significant interactions between pairs of scales. Interaction effects between scales are evidenced by observing how the behavior of rank diversity depends on the specific values of other scales as one scale increases. Since such interaction effects may be subtle to discern graphically, a statistical approach serves to validate the hypothesis of their existence.

These models were fitted using the $\log_{10}$ scaled values of temporal and grammatical scales as predictors and μ as a response. In particular, the multiplicative model is
(6)$$Y=\beta_{0}+\beta_{1}X_{1}+\beta_{2}X_{2}+\beta_{3}X_{3}+\beta_{4}X_{1}X_{2}+\beta_{5}X_{1}X_{3}+\beta_{6}X_{2}X_{3}.$$
The model is simplified into a linear form by excluding terms involving products of predictors ($X_{i}$). Here, the coefficients ($\beta_{i}$) represent weights determining the influence of each predictor (i.e., a particular scale) on the response variable, μ. Hypothesis tests are employed to ascertain whether a coefficient significantly deviates from zero, indicating evidence of influence. In cases involving products of two scales, the coefficients quantify the impact of one scale on how the other affects the response. This is evident by factoring a common linear predictor, such as $X_{1}$: $Y=\beta_{1}X_{1}+\beta_{4}X_{1}X_{2}+\beta_{5}X_{1}X_{3}=\left(\beta_{1}+\beta_{4}X_{2}+\beta_{5}X_{3}\right)X_{1}=\beta_{145}\left(X_{2},X_{3}\right)X_{1}$, where $\beta_{145}\left(X_{2},X_{3}\right)$ serves as a multivariate function slope of $X_{1}$, delineating how $X_{2}$ and $X_{3}$ influence the relationship between $X_{1}$ and *Y*. Statistically significant coefficients $\beta_{4}$ and $\beta_{5}$ indicate pairwise interaction between predictors $X_{2}$ and $X_{3}$ with $X_{1}$. Geometrically, $\beta_{0}$ corresponds to the level of the hyperplane best fitting the observations; however, it does not provide further pertinent information in this context. All models were fitted using linear least-squares problems solved via the QR factorization method for numerical stability, leveraging the standard lm function in the R programming language.

## 4. Results and Discussion

The rank diversity of *N*-grams is computed for tweets originating from eight distinct countries, which were evenly divided between Spanish-speaking nations (Mexico, Argentina, Colombia, Spain) and English-speaking nations (Canada, United Kingdom, India, South Africa). Here, Δt represents the duration between consecutive time “slices” with the total time span denoted as TΔt. In Figure 5, we present the rank diversity plots for *N*-grams, employing a time interval of Δt=24 h across varying values of *N*, ranging from 1 to 5.

Initially, we observe that the sigmoid curve remains a suitable model for describing rank diversity patterns, which is particularly evident when examining shorter time intervals. This consistency holds across all combinations of scales considered. Furthermore, for N=1, the rank diversity profiles exhibit substantial similarity. However, as we increase *N*, the profiles begin to diverge. This discrepancy implies that 1-grams demonstrate consistent rank diversity regardless of language or country. In contrast, for 2-grams, 3-grams, and 4-grams, distinct trends emerge. Specifically, the curves for Spanish-speaking countries exhibit close alignment, forming a cohesive cluster, while Canada displays a noticeable deviation from this pattern.

In Figure 5, the Spanish-speaking countries appear to be more tightly clustered in general than the other countries studied. Therefore, the rank diversity differences among the Spanish-speaking countries are smaller than the other non-Spanish speaking countries. We calculated this difference to strengthen this argument by using the least squares difference for all countries against all others. In Table 1, we can see that the difference among Spanish speakers ranges from 0.001 to 0.004, while it ranges from 0.005 to 0.21 for other countries. We are unsure if the differences are due to English itself (as India is very similar to Spanish-speaking countries) or their secondary languages that have words in the ranks considered (French in Canada, Hindi and others in India, Afrikaans and others in South Africa). It would be necessary to analyze what percentage of words in different ranges come from other languages and possibly other factors, but this is beyond the scope of the article.

This behavior means that there are some other features that make them distinct from the rest.

In English-speaking countries like the UK, Canada, and India, second languages play a significant role, often reflecting the cultural differences. Canada is officially bilingual, with both English and French as official languages. In the UK, the most commonly learned second languages are French, Spanish, and German. India is a highly multilingual country with English serving as an associate official language alongside Hindi. In each region, that should be checked, but this would require an anthropological study, which is way beyond the statistical linguistics scope of this paper.

Sigmoid curves can be likened to the cumulative distribution of a Gaussian with μ denoting the mean (the peak and center of the Gaussian) and σ representing the standard deviation (which determines its width). Within the sigmoid context, μ denotes the *x*-axis point where the range diversity equals 0.5, while sigma governs the smoothness of the sigmoid (if it were zero, it would resemble a step from zero to one at μ; as it increases, the sigmoid widens). In a previous study ([29]), we conducted a comparative analysis of various grammatical scales across six languages using Google Books data. Typically, σ exhibits a correlation with μ, although it has higher variability. Consequently, we streamlined the range diversity curves, focusing solely on μ for comparisons in this article.

In the subsequent sections, we delve into the impacts of different scales (grammatical, spatial, and temporal) on rank diversity, leveraging estimates of the parameter μ from the sigmoid curves.

### 4.1. Grammatical Scale

Following the progression of *N* values in ascending order, Figure 6 and Figure 7 illustrate that as the grammatical scale expands, so does the pace of rank diversity increment. This trend holds consistently across countries, regardless of whether the language is Spanish or English, or the values adopted by the other two scales. Notably, a larger grammatical scale signifies an increase in phrase complexity. At the apex of the scale (5-grams), the rank diversity pertains to blocks comprising five words. The potential combinations of five-word blocks surpass those of four words, which in turn exceed those of three words. Consequently, we observe heightened diversity at the initial ranks compared to lower grammatical scales.

Furthermore, we observe that for 1-grams, μ exhibits similarity between Spanish and English, remaining practically unaffected by the spatial scale. In other words, there are no discernible changes in the pace of rank diversity increment across analyzed areas. However, it does fluctuate in relation to the temporal scale, as evidenced by the findings in the first column of Figure 6. This underscores the significance of employing diverse scales to scrutinize the rank diversity of languages.

Additionally, a visual inspection reveals that alterations in the grammatical scale yield the most significant overall escalation in the pace of rank diversity increment compared to the other two scales. Toward the conclusion of the Results section, we quantitatively validate this observation by comparing average dispersions.

### 4.2. Temporal Scale

In Figure 6, we manipulate the temporal scale along the *x*-axis to illustrate the correlation between μ and various time intervals Δt. Notably, we observe that the pace of the rank diversity increment does not exhibit a linear increase as observed in the grammatical or spatial scales; instead, it displays a noticeable concave shape. This nonlinear effect stems from the phenomenon whereby the addition of frequencies results in fewer variable positions for the *N*-grams in the lists constituting the total timespan under analysis. Consequently, μ experiences an increment until reaching a certain time interval, after which it begins to decline. This phenomenon occurs because the denominator in the calculation of rank diversity, which represents the number of possible lists dividing the total timespan, decreases with higher temporal scales. Consequently, the relationship between the speed of rank diversity increment and the temporal scale remains consistent regardless of the country or language. Additionally, it is noteworthy how the shape of the relationship between μ and time alters across various columns, each representing different grammatical scales. This observation suggests that the grammatical scale exerts an influence on the variation of the temporal scale.

### 4.3. Spatial Scale

In each plot of Figure 6, particularly from the column N=3 onwards, it is evident that with fixed grammatical and temporal scales, the spatial scale influences the speed of rank diversity increments. However, to elucidate this relationship further, Figure 7 plots μ against the spatial scale. Notably, for Spanish-speaking countries, μ decreases with the spatial scale when the grammatical scale exceeds 1, whereas in general, μ remains relatively unchanged against the spatial scale for the United Kingdom. Further investigation is necessary to explore potential explanations, such as whether there exists greater homogeneity in the United Kingdom compared to the other countries and/or whether these results reflect a distinction between English and Spanish. Overall, it is observed that μ also decreases with the grammatical scale.

### 4.4. Relevance of Scales

The interdependence of different types of scales is evident in their overlapping influences, indicating their lack of independence. A detailed explanation follows.

Now, we address the question of which scale holds the most significance in terms of its impact on the variability of μ and, consequently, on the behavior of rank diversity itself.

To investigate this, we assess the relative importance of these scales using Equations (Equation 3)–(Equation 5). The results, depicted in Figure 8, confirm that the grammatical scale exhibits the greatest variance relative to the other scales under consideration. Additionally, for all Spanish-speaking countries, both the temporal and spatial scales appear to hold approximately equal importance. However, in the case of the United Kingdom, despite having more data available, the spatial scale seems to possess less significance.

Finally, we present the *p*-values and the associated estimated *F*-statistic for the first four terms of model (Equation 6) in Table 2.

The significance of the observed difference increases as the *p*-value decreases. Since our results yield a *p*-value lower than 0.05, we consider it statistically significant. These low *p*-values indicate that at least one scale is linked to μ, assuming approximate linear correlation.

Our main objective here was to support the hypothesis that changes in certain scales lead to variations in rank diversity behavior. In other words, we aimed to demonstrate that the linear regression model offers a better fit to the data than a model with no independent variables, which implies no influence of the scales on the observed variability.

Specifically, we tested the significance of each associated coefficient to determine whether a particular scale is related to μ. The resulting *p*-values are detailed in Table 3.

It is noteworthy that for the United Kingdom, the temporal and spatial scales exhibit les significance according to our test compared to the grammatical scale. As illustrated previously in Figure 7, the spatial scale appears to be practically independent of μ. However, in the case of the temporal scale, this suggests that a linear approximation alone is insufficient to adequately capture the relationship between these scales and μ.

Therefore, employing a quadratic model proves to be more effective in elucidating the existence of relationships for the temporal scale. This approach reveals that for this dataset, the temporal relation with μ is nonlinear, as evidenced by the observations depicted in Figure 6.

Alternatively, to assess the statistical significance of interactions between pairs of scales, we can examine the *p*-values associated with *t*-tests conducted on the estimated coefficients $\beta_{4}$, $\beta_{5}$, and $\beta_{6}$ in model Equation 6. The results are presented in Table 4.

We observe that all interactions between the grammatical and spatial scales, as well as between the grammatical and temporal scales, are statistically significant. However, the significance is somewhat lower for Argentina. Additionally, there is a notably higher level of interaction between the grammatical and spatial scales, specifically for the United Kingdom.

It is interesting that we found minor statistical differences between English and Spanish (compared to other variables). Most of the linguistic studies comparing these two languages focus on other aspects such as phonetics [37]. It would be relevant to make further statistical comparisons across languages.

### 4.5. Special Tokens

In this section, we concentrate on examining special tokens commonly employed in Twitter discourse: emojis [38,39], hashtags [40], and mentions [41,42]. We investigate the most prevalent occurrences of each within Argentina, Mexico, Spain, and the United Kingdom, along with their respective rank diversities.

An emoji is a pictogram, logogram, ideogram, or smiley utilized in electronic messages and web content. Emojis serve the primary purpose of conveying emotional nuances that may be absent in typed communication. They represent images that can be depicted as encoded characters. Given their widespread usage, emojis play a pivotal role in enhancing our online communication by supplementing text with emotional context akin to body language and facial expressions.

Figure 9 presents a comprehensive overview of the most frequently used emojis, which are arranged in descending order from top to bottom. Notably, the *Smiling Face with Heart-Shaped Eyes* and *Face with Tears of Joy* emojis emerge as the predominant expressions, which are indicative of sentiments like happiness, affection, and amusement. These are followed by a diverse array of symbols representing various emotions, such as love, strength, and positivity.

Interestingly, while some emojis exhibit universal popularity across all countries, others demonstrate regional prevalence (consistent with the results of [38]). For instance, the *Woman Dancing* and *Sun with Face* emojis are predominantly used in Spain, whereas the *Sleeping Face* emoji is more prevalent in Mexico, and the *Unamused Face* emoji is favored in Argentina. Conversely, the *Fire* emoji appears to be exclusive to the United Kingdom.

Furthermore, among the most frequently used Unicode emoji symbols, there are instances of *Emoji Modifier Fitzpatrick*. These emojis can be customized to reflect different skin tones using one of five modifiers. Notably, Argentina’s top list includes only the *Light Skin Tone* modifier, while Spain also incorporates the *Medium-Light Skin Tone*. In contrast, both Mexico and the UK feature the *Medium Skin Tone* modifier, with neither country including the *Medium-Dark Skin Tone* or *Dark Skin Tone* in their lists.

A *hashtag* serves as a metadata tag commonly employed on platforms such as Twitter and Instagram. Recognizable by the prefix of the hash symbol *#*, hashtags function as user-generated labels facilitating the organization and cross-referencing of content based on specific topics or themes. Users leverage hashtags to discover and engage with content relevant to their interests, as the practice enables the aggregation of posts related to a particular hashtag.

It is pertinent to emphasize that hashtags lack centralized registration or control, allowing any user to create or use them freely. This decentralized nature means that hashtags do not adhere to predefined definitions, and their significance can evolve over time. Consequently, a single hashtag may serve multiple purposes, and the interpretation or accepted meaning of a hashtag is subject to change within the context of online discourse.

Table 5 reveals that the hashtag *#trndnl* emerges as the predominant hashtag across the dataset, which is indicative of discussions surrounding popularity and trending topics. Additionally, notable cities within each country are prominently featured, such as Buenos Aires, Cordoba, and Rosario in Argentina; CDMX/Mexico City, Monterrey, Guadalajara, and Puebla in Mexico; Madrid, Barcelona, and Sevilla in Spain; and London and St Albans in the UK. An intriguing observation is the prevalence of weather-related hashtags exclusively within the European context.

A *mention* is a tweet that contains another person’s username anywhere in the body of the tweet. User mentions are identified with the @ symbol within tweets.

Table 6 provides insight into the most frequently mentioned users on Twitter, shedding light on various meeting places such as shopping centers (e.g., @galeriasmx in Mexico), cinemas (e.g., @cinemex in Mexico), and airports (e.g., @heathrowairport in the United Kingdom), as well as renowned companies (e.g., @starbucksuk) and individuals such as politicians (e.g., @mauriciomacri in Argentina) and artists (e.g., @officialmaki in Spain). Notable differences can be discerned across countries, which are indicative of divergent Twitter usage patterns during the period under examination. For instance, Argentina exhibits numerous mentions of artists, Mexico features several references to commercial franchises, Spain includes soccer teams and political parties, while the United Kingdom predominantly mentions the National Rail network.

It is imperative to acknowledge that these mentions are derived from geolocated tweets, representing only a fraction of all tweets. Consequently, these mentions may exhibit bias, and the most popular accounts may differ from those enumerated herein.

Rank diversity curves pertaining to emojis, hashtags, and user mentions can be effectively approximated by sigmoid curves as seen in Figure 10, which is similar to many other phenomena [34]. Across all cases, user mentions emerge as the most diverse feature per country, while emojis exhibit the lowest diversity.

## 5. Conclusions and Future Work

In conclusion, our analysis of geolocated Twitter data has provided valuable insights into language usage across different spatial, temporal, and grammatical scales.

Our findings indicate a clear relationship between higher scales and elevated values across the studied dimensions. For instance, monograms represent the lowest scale in the grammatical hierarchy, while 5-grams signify the highest. Similarly, spatial scales ranging from 3 to 3000 km, and temporal scales from 3 to 96 h, reflect the continuum of spatial and temporal granularity.

Interestingly, while higher scales generally correlate with greater rank diversity, we observed a nuanced trend in the temporal dimension. Specifically, temporal rank diversity exhibited a concave pattern with both shorter and longer time intervals demonstrating higher diversity compared to intermediate intervals.

Assessing the relative importance of each scale based on rank diversity dispersion, we identified the grammatical scale as the most influential among the three. Moreover, we observed similar levels of importance for temporal and spatial scales in Spanish-speaking countries, whereas the spatial scale emerged as less critical in English-speaking countries within the scope of the statistical measures employed.

Looking ahead, future research could delve deeper into understanding the complex interactions between these scales and their impact on language dynamics, potentially exploring additional linguistic features and employing more sophisticated analytical techniques. Additionally, investigating longitudinal trends and incorporating sociocultural factors may offer further insights into the evolving nature of language use in online platforms like Twitter.

Indeed, variations in rank diversity on Twitter are likely influenced by a multitude of factors, including the topics of interest among users in each country. For instance, countries experiencing significant economic or political upheavals may witness a narrower range of topics dominating discussions, leading to lower diversity in word usage. Conversely, the occurrence of events over shorter durations or the overlapping of multiple events could contribute to higher diversity in rank. This pattern of behavior sheds light on the observed differences across geographical areas, such as tweets originating from urban centers versus those from rural areas. Furthermore, it could also explain the differences in the usage of hashtags and mentions.

This underscores the complexity of the relationship between the considered scales and rank diversity. While the grammatical scale emerges as strongly related to changes in the speed of rank diversity increment, a comprehensive understanding of this relationship requires the consideration of multiple scales simultaneously. Therefore, future research should explore the intricate interplay between spatial, temporal, and grammatical scales to unravel the nuanced dynamics of language usage on Twitter comprehensively.

The consistent fit of the sigmoid curve to rank diversity curves across various spatial and temporal scales is intriguing. It suggests that the underlying mechanisms shaping rank diversity remain unaffected by changes in language or scales. This finding not only extends our previous research, which focused on temporal scales of years, but also underscores the robustness of the sigmoid curve in capturing language dynamics across different temporal and geographical contexts. Notably, while the diversity of monograms remains unaffected by spatial scale, higher grammatical scales exhibit varying rank diversity patterns across different spatial scales, implying greater language use variability at higher grammatical scales.

Our examination of the most frequent emojis across different languages and countries reveals intriguing cultural differences among Twitter users. Emojis, as nonverbal symbols, not only reflect cultural nuances but also potentially convey collective sentiment and biases within each country. Similarly, hashtags, as embedded metadata, play a significant role in fostering communities, evoking emotions, and highlighting relevant topics and events within a community. Understanding the dynamics of hashtags can offer valuable insights into societal trends and preferences. Furthermore, the analysis of mentions in Twitter unveils the changing relevance of various entities over time and space, reflecting evolving trends and interests.

The COVID-19 pandemic has brought to the forefront the issue of misinformation dissemination through social media platforms, prompting academic research to scrutinize its impact. Studies, such as the one by Pennycook et al. [22], have shed light on the disconnect between individuals’ assessments of news accuracy and their intentions to share it, suggesting that misinformation may proliferate even when not explicitly endorsed. This underscores the importance of prioritizing accuracy to mitigate the spread of misinformation online.

As highlighted by the Pew Research Center [43], a significant portion of the population, including approximately 72 percent of Americans, relies on social media platforms for news consumption. However, the inherent tendency of false information to spread more rapidly than factual news on platforms like Twitter underscores the urgency of investigating statistical linguistics within the specific context of misinformation dissemination.

Therefore, a natural extension of current research efforts would involve delving deeper into the linguistic patterns and dynamics associated with the propagation of misinformation on social networks. By analyzing linguistic features such as vocabulary usage, grammatical structures, and temporal patterns, researchers can gain valuable insights into the mechanisms driving the dissemination of false information on social media platforms. Such insights can inform the development of strategies and interventions aimed at curbing the spread of misinformation and promoting the dissemination of accurate and reliable information during public health crises and beyond.

Recently, large language models (LLMs) have applied available data to various artificial intelligence (A.I.) [44] applications. Studying the range diversity in human-generated and LLM-generated text could help distinguish between them.

## Figures and Tables

**Figure 1 entropy-26-00734-f001:**
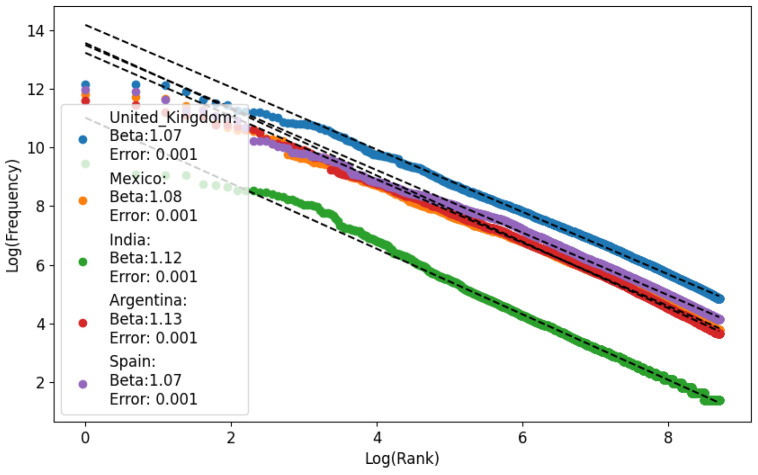
Zipf analysis of monogram data. Errors less than 0.05 show a good fit with Zipf’s law, although models with more variables can produce better fits [30]. In texts with beta exponents between 1<β<2, the variety of lexical units employed is restricted, and there are many repetitions. Dotted lines show the lineal regression of Zipf’s beta exponent.

**Figure 2 entropy-26-00734-f002:**
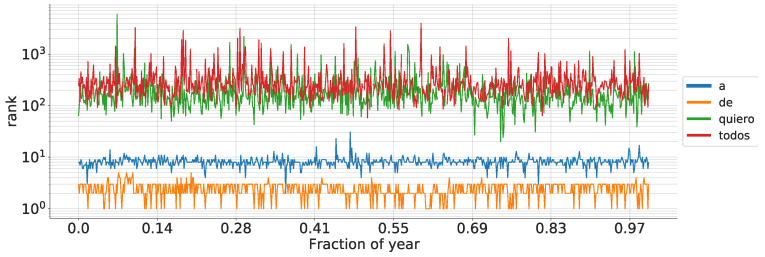
Rank development of some Spanish words during 2014 using a temporal scale of 24 h. Note that if a word has a high rank, its trajectory takes a wider set of possible ranks compared to words with lower ranks [28].

**Figure 3 entropy-26-00734-f003:**
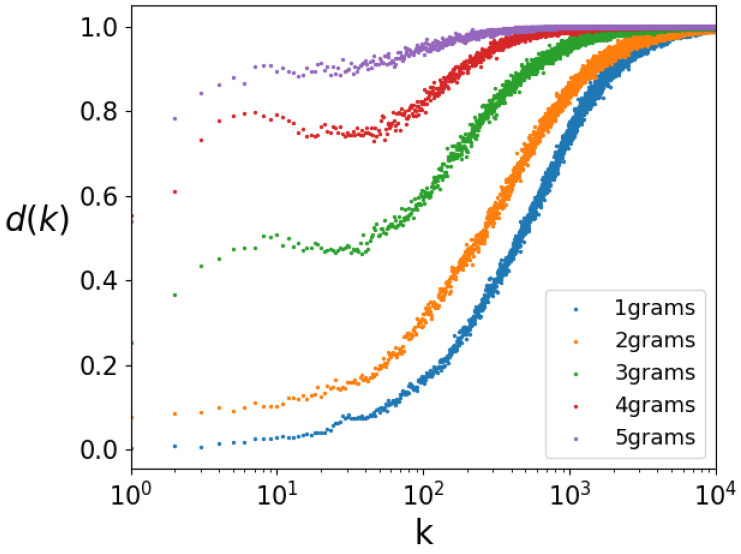
Sigmoid curves for 1, 2, 3, 4, and 5-grams showing that Zipf distribution is valid for a period of 3 h (smallest temporal interval).

**Figure 4 entropy-26-00734-f004:**
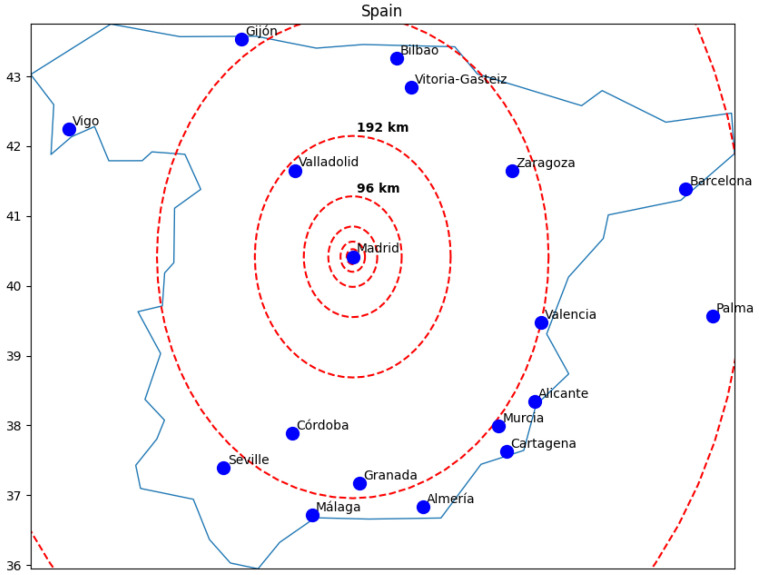
Concentric circles illustrating the geographic distribution for spatial scale in Spain.

**Figure 5 entropy-26-00734-f005:**
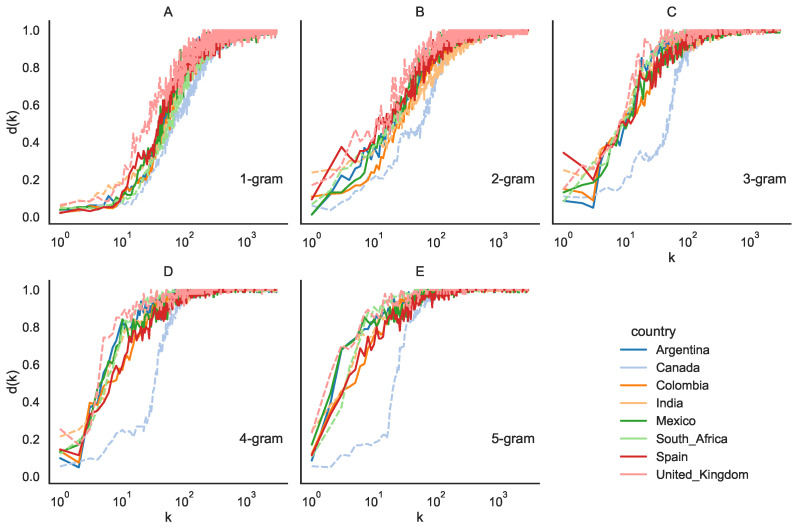
Rank diversity d(k) for eight different countries at different grammatical scales N=1,2,…,5, Δt=24 h. Spanish-speaking countries are shown with solid lines, English-speaking countries are shown with dashed lines. (**A**) 1-grams, (**B**) 2-grams, (**C**) 3-grams, (**D**) 4-grams, and (**E**) 5-grams. X-axis is shown in log scale.

**Figure 6 entropy-26-00734-f006:**
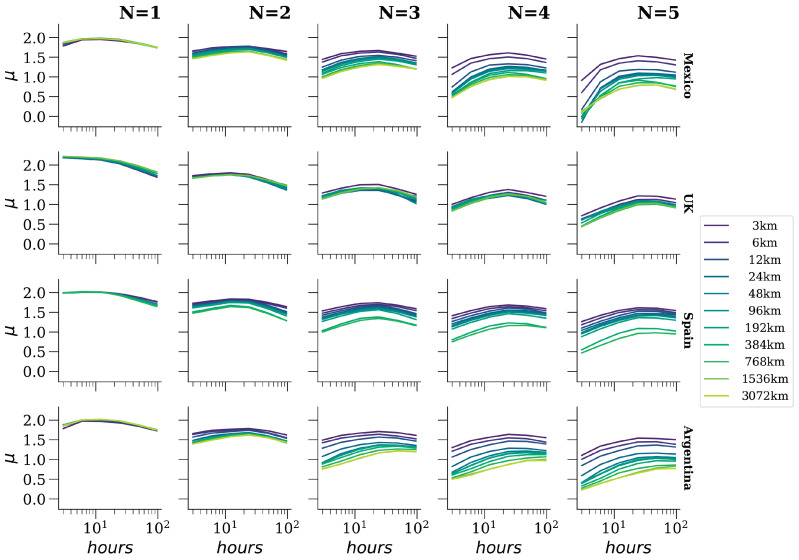
Different estimations of μ. Measure the speed of rank diversity increment vs. different values of the temporal scale. Note that the *x*-axis is in $\log_{10}$ scale. The *N* values represent *N*-grams (grammatical scale). Each row uses Twitter data from the country specified at the right label. Colored lines represent the different spatial scales considered.

**Figure 7 entropy-26-00734-f007:**
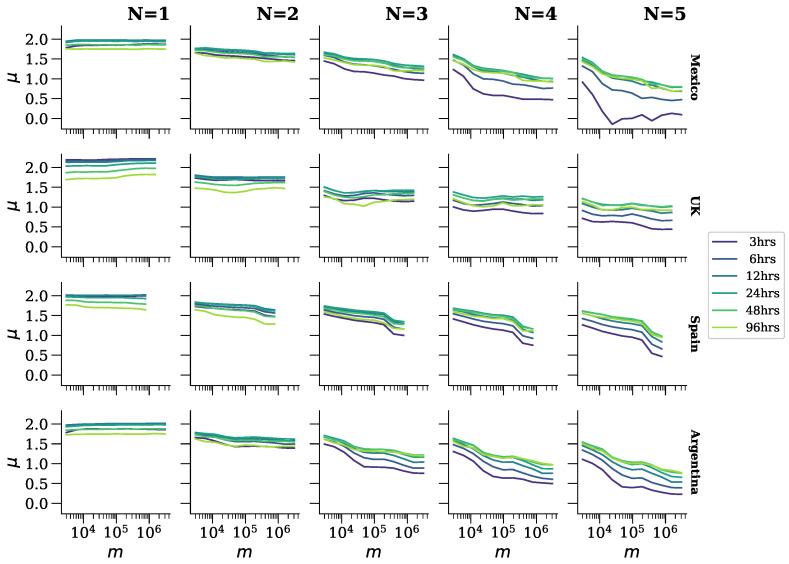
μ estimates vs. different values of the spatial scale. The top and right labels represent the same as in Figure 6. Colored lines here represent the different temporal scales.

**Figure 8 entropy-26-00734-f008:**
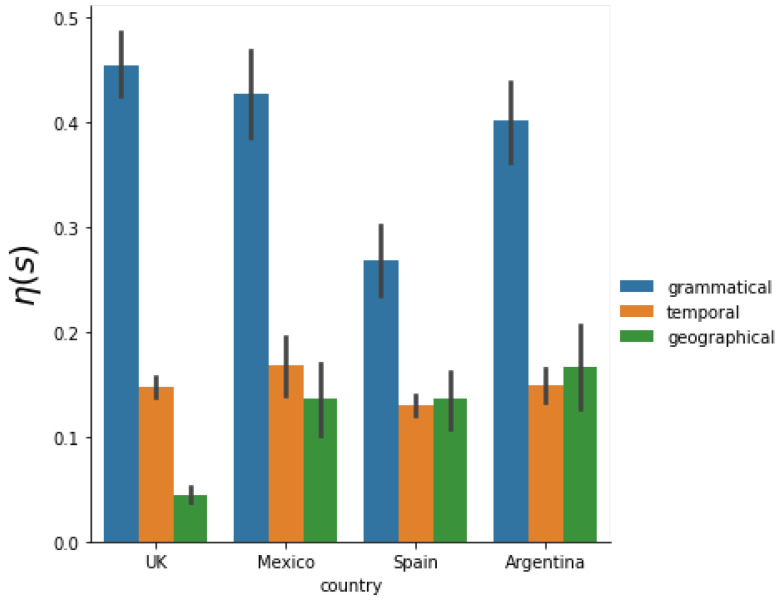
Average dispersion (2) of scale *s* for each country and a bar chart to compare them.

**Figure 9 entropy-26-00734-f009:**
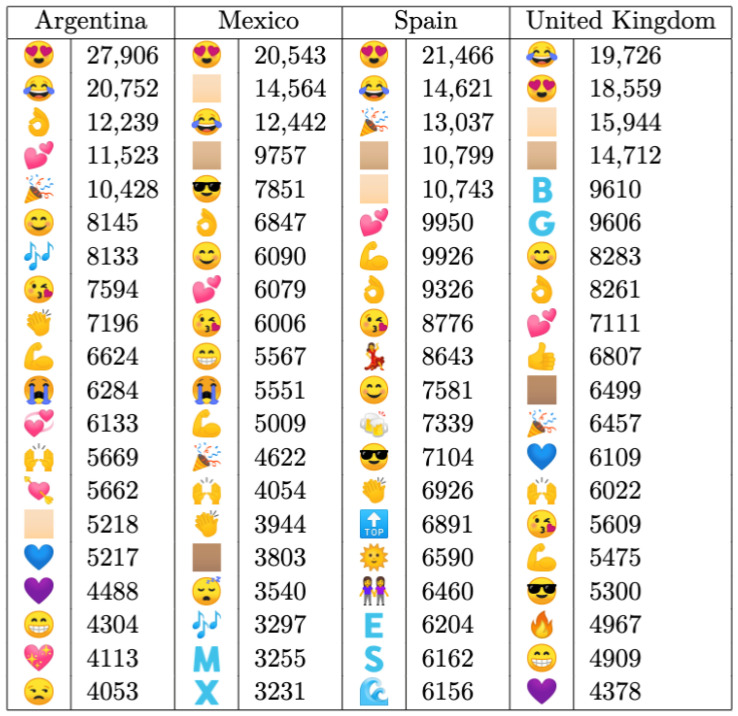
Comparison of the most frequent emojis and modifiers shown in Argentina, Mexico, Spain and the United Kingdom in geolocalized tweets from the year 2014. The number listed is the frequency of the emojis.

**Figure 10 entropy-26-00734-f010:**
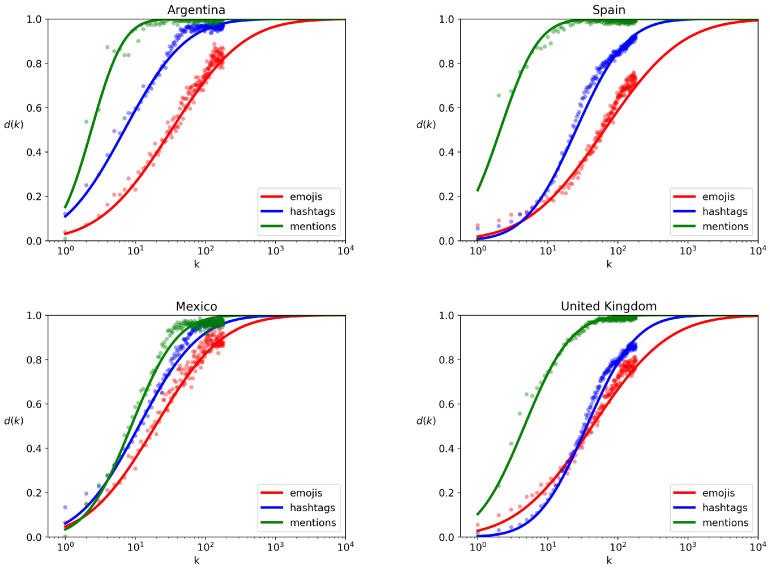
Rank diversity d(k) of emojis, hashtags and mentions in Argentina, Mexico, Spain and United Kingdom in a time interval Δt of 24 h. Dots represent the rank diversity, and solid curves represent the sigmoid approximation.

**Table 1 entropy-26-00734-t001:** Distance between countries using least square difference.

	Mexico	Colombia	Spain	United Kingdom	Canada	South Africa	India
Argentina	−0.003	0.002	−0.001	−0.007	0.014	0.006	0.001
Mexico		0.004	0.002	−0.005	0.017	0.008	0.003
Colombia			−0.002	−0.009	0.012	0.004	−0.001
Spain				−0.007	0.015	0.006	0.002
United Kingdom					0.021	0.013	0.008
Canada						−0.008	−0.013
South Africa							−0.005

**Table 2 entropy-26-00734-t002:** *p*-values associated with the *F*-statistic, which represents the significance of the regression, i.e., that at least one scale is contributing to explain the changes in rank diversity. The parameter *l* is the number of parameters used and *H* is the number of observations.

Country	Mexico	United Kingdom	Argentina	Spain
l−1	3	3	3	3
H−l	326	296	326	266
*F*-statistic	489.6	631.7	546	261
*p*-value	<$2.2\times 10^{-16}$	<$2.2\times 10^{-16}$	<$2.2\times 10^{-16}$	<$2.2\times 10^{-16}$

**Table 3 entropy-26-00734-t003:** Associated *p*-values to *t*-tests of the linear coefficients. $X_{1}$,$X_{2}$, and $X_{3}$ represent the grammatical, spatial, and temporal scales, respectively.

*p*-Value	$X_{1}$	$X_{2}$	$X_{3}$
Mexico	<$2.0\times 10^{-16}$	<$2.0\times 10^{-16}$	$9.8\times 10^{-16}$
United Kingdom	<$2.0\times 10^{-16}$	0.125	0.426
Argentina	<$2.0\times 10^{-16}$	<$2.0\times 10^{-16}$	$<2.0\times 10^{-16}$
Spain	<$2.0\times 10^{-16}$	<$2.0\times 10^{-16}$	0.0183

**Table 4 entropy-26-00734-t004:** Associated *p*-values to *t*-tests of the interaction coefficients. $X_{1}$, $X_{2}$, and $X_{3}$ represent the grammatical, spatial, and temporal scales, respectively.

*p*-Value	$X_{1}$ * $X_{2}$	$X_{1}$ * $X_{3}$	$X_{2}$ * $X_{3}$
Mexico	<$2.2\times 10^{-16}$	<$2.2\times 10^{-16}$	0.57
United Kingdom	<$8.2\times 10^{-5}$	<$2.2\times 10^{-16}$	0.12
Argentina	<$2.2\times 10^{-16}$	<$2.2\times 10^{-16}$	<$2.6\times 10^{-4}$
Spain	<$2.2\times 10^{-16}$	<$2.2\times 10^{-16}$	0.47

**Table 5 entropy-26-00734-t005:** Most frequent hashtags in Argentina, Mexico, Spain and United Kingdom in 2014 geolocalized tweets. The number listed is the frequency of the hashtags.

Argentina		Mexico	
#trndnl	9518	#trndnl	10,836
#buenosaires	5022	#cdmx	6951
#argentina	2372	#mexico	6540
#me	1646	#mexicocity	3631
#cordoba	1502	#job	3508
#rosario	1419	#hiring	3135
#love	1269	#monterrey	2870
#selfie	1088	#endomondo	2626
#friends	1069	#guadalajara	2433
#job	961	#endorphins	2299
#endomondo	939	#friends	2049
#viernes	911	#méxico	1860
#amigos	891	#love	1758
#repost	873	#careerarc	1575
#hiring	868	#photo	1486
#night	807	#jobs	1370
#endorphins	804	#quieremeamame	1333
#domingo	774	#puebla	1215
#sabado	727	#selfie	1183
#carlosrivera	719	#travel	1069
**Spain**		**United Kingdom**	
#trndnl	19,328	#nowplaying	52,085
#madrid	14,552	#london	35,934
#barcelona	14,167	#job	25,486
#incdgt	13,367	#tnc	24,897
#dgt	12,978	#trndnl	24,319
#meteocat	12,260	#areacode	23,084
#endomondo	9689	#hiring	22,587
#meteo	8736	#tides	20,236
#spain	8649	#ktt	19,237
#blanco	8644	#weather	12,024
#endorphins	8559	#careerarc	11,841
#324meteo	8468	#essex	11,016
#meteo	3127	#broadbandcompareuk	10,881
#retención	8189	#bestbroadband	10,641
#precaución	7798	#photo	10,492
#elcatllar	6844	#ukweather	9978
#obra	5398	#endomondo	9770
#arameteo	4910	#jobs	9375
#sevilla	4890	#endorphins	8988
#amarillo	4030	#stalbans	7498

**Table 6 entropy-26-00734-t006:** The most frequent users mentioned in Argentina, Mexico, Spain, and the United Kingdom in geolocalized tweets from the year 2014. The number listed is the frequency of the mentions, and the type column explains the class account. Existing personal accounts are anonymized.

Argentina			Mexico		
**Account**	**Frequency**	**Type**	**Account**	**Frequency**	**Type**
@clubsolotu_arg	724	Fans Club	@aicm3	3965	Airport
@vale975	649	Radio Channel	@cinemex	2629	Cinema
Personal account	543	Weather Technician	@cinepolis	2352	Cinema
@rkartista	432	Artist	@germanmontero5	2296	Singer
@aa2000oficial	417	Argentina Airports	@grupointocable	2122	Musical Group
@mauriciomacri	368	President of Argentina	@sonadoraeterno	2108	Fan Club
@abrahammateomus	339	Artist	@mariobautista_	1878	Musical Artist
@lucianopereyra	325	Singer	@smartfit_mex	1755	Gym
@todonoticias	263	News Channel	@walmartmexico	1274	Supermarket
@c5n	243	News Agency	@galeriasmx	1010	Shopping Center
@infobae	179	News Agency	@chilismexico	789	Restaurant
@radiomitre	169	Radio Station	@aeropuertodemty	729	Airport
@rialjorge	163	Journalist	@aeropuertosgap	695	Airport
@cfkargentina	156	VP of Argentina	@solosanborns	673	Store
@marialuizateodo	153	Deleted Account	@banamex	654	Bank
@starbucksar	153	Coffee Shop	@lacasadetono	644	Restaurant
@niallofficial	152	Singer	@oasis_coyoacan	625	Shopping Center
@sole_pastorutti	152	Artist	@auditoriomx	594	Entertainment Center
@brigitte2300	144	Singer	@perisur	566	Shopping Center
@lanacion	144	Newspaper	@tuado	559	Bus Company
**Spain**			**United Kingdom**		
**Account**	**Frequency**	**Type**	**Account**	**Frequency**	**Type**
@canalfiesta	1124	Radio Channel	@nationalrailenq	4797	National Rail
@aena	1068	Airports Management	@heathrowairport	2088	Airport
@aenaaeropuertos	842	Airport	@starbucksuk	1480	Coffee Shop
@dominguezja	830	Media Personality	@luvthenorth444	1401	Suspended Account
@oficialmaki	671	Compositor	@brewdog	1030	Brew Restaurant
@oficiallamorena	570	Singer	@simmons2k	928	Politician
@adif_es	462	Railway Management	Personal account	903	Chelsea Fan
@willylevy29	395	Actor	@costacoffee	902	Coffee Shop
@realmadrid	321	Football Club	Personal account	863	Chelsea Fan
@pinedademar	297	City Hall	@babs200475	819	Suspended Account
Personal account	288	William Levy’s Fans	@shelleym1974	774	Suspended Account
@pablo_iglesias_	278	Politician	@luvyorkshire444	693	Suspended Account
@sanchezcastejon	273	Political leader	Personal account	659	Chelsea Fan
@psoe	265	Political Party	Personal account	634	Woman
@fcbarcelona	242	Football Team	@harrods	625	Department Store
@barcelona_cat	234	Barcelona News	Personal account	576	Chelsea Fan
@ahorapodemos	213	Political Party	Personal account	560	Chelsea Fan
@renfe	202	Railway Transportation	@selfridges	560	Department Store
@el_pais	199	Newspaper	@visitlondon	481	Travel Guide
Personal account	190	Fan Club Manager	@skynews	466	News Channel

## Data Availability

The data used for simulations in this study are available on request from the corresponding author.

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
