# Peer review of "Language Statistics at Different Spatial, Temporal, and Grammatical Scales"

_entropy, 2024, doi:10.3390/e26090734_

Round 1

Reviewer 1 Report

Comments and Suggestions for Authors

The submission deals with a relevant and challenging topic. The authors try to observe language change over short time intervals.

The reason why this is interesting is not give. Is that helpful to explore geo or temporal language variants?

The data follows a Zipf distribution. Therefore, it would be best to map all distributions as Zipf-distributions and compare the different exponents. The diversity in languages regarding the exponent might be another way to look at the same thing. Therefore, the exploration with other metrics is necessary.

Rank diversity is the main metric that is explored. It is strange that language models or different rankings are not compared with any other metric. Rank diversity actually gives very little information about the distribution. More adequate and established methods would be to compare the language models (e.g. with KL divergence) or rank correlation coefficients.

A method would be to compare all rankings to the overall ranking for the entire corpus.

A convincing reason to use rank diversity and to use 5-grams are not given.

We do not get any information about the amount of data in the “smallest” unit (3 hour period for a region with few tweets). Is that still sufficient to compare to the “real” distribution? It would be necessary to check whether the smallest models still are valid Zipf distributions for the respective language.

A figure to illustrate how geographic distribution was modelled would be helpful. Is this modelled as a expanding circle? Close to a center of far away without considering how the other cities are distributed? Maybe the form of the UK explains some of the results in figure 5?

Some results on emojis and hashtags show the most frequent entities. However, the method is aiming at detecting changes. What are the entities which exhibit the strongest changes? These would be emojis or hashtags that are popular at some time or at some location but not at others.

Furthermore, address the fraction of tweets including emojis etc. and check whether this changing with time or region.

Abstract need to mention:

-          the paper talks about word-ngrams (not character ngrams)

-          time period analyzed

-          different countries were analyzed

Looking at a temporal distribution as the fraction of a year is uncommon.

Several figures are too small. Also tabela 5 is too tiny. Reformat.

The state of the art section is rather sketchy and “hidden” in the introduction. It should be a proper section. Connect your work with previous work on language models.

Check previous publications on differences between Spanish and English variants and show how your work relates to them regarding method and results.  

Also work on emojis should be discussed and related to the results of the paper. Check e.g. https://aclanthology.org/W16-2610.pdf

Pentagram is mostly used for a star. Better refer to 4-grams and 5-grams.

The terminology seems not adequate. Looking at n-grams of words is not a grammatical dimension of language. It would be better called lexical dimension. Also the words scale is not really the best. Needs to be changed throughout the paper.

References: last reference incomplete

Comments on the Quality of English Language

Details:

potential disparities -> unclear

greatest dispersion?

Rysunek?

Reviewer 2 Report

Comments and Suggestions for Authors

The authors present a study of so called rank diversity in language extracted from Twitter, and in particular they try to determine which geographical scale, temporal scale or grammatical scale is the most important contribution to rank diversity.

While I believe that studies of general statistical patterns in language are important to our understanding of language use, this work has, in my opinion, some significant flaws which make it difficult to assess the value of the results.

1. The work is mostly concerned with understanding what factors contribute to rank diversity, and the conclusions are mostly about this. They do not establish *why* understanding rank diversity is important in the first place: what did we learn about language and language use form this study? If one factor contributes more to rank diversity than another, what does this allow us to do or to understand that we couldn't before?

2. More fundamentally, the reliability of rank diversity as a measure is not established. Being a statistical sampling measure, fluctuations in which object occupies which rank are expected, and indeed must increase as rank increases, as the number of examples of items in the sample decreases as rank increases. We should also expect significant sample size effects.

The authors do not study to what extent their results stem simply from sampling effects, and in particular, how much of the variation in rank diversity can be explained by variations in the sample sizes.

Instead, samples of greatly different sizes are compared side by side, both between "scales" and between different datasets.

Without properly understanding these sampling effects, it is impossible to determine whether the results presented are significant or not.

One standard way to do this would be to compare with a null model, randomly sampling from a Zipf type ranked distribution, and observing what values of rank diversity are to be expected from samples and populations matching the analyzed data. Then one can see how much (if at all) the results from data differ from this statistical baseline.

Besides these two main points, I have the following more minor comments:

a. Some of the interpretation of results is questionable. For example, on page 6, lines 232-33, the authors note that the Canada data diverges from a cluster of Spanish-speaking countries. But the curves for India and the UK both largely follow the Spanish speaking countries. This suggest that there is something specifically different about the Canada data, and it has nothing to do with not speaking Spanish.

b. The discussion of mu and eta (page 5), is difficult to follow, and could be unpacked somewhat.

I cannot recommend publication of this article until the two main flaws I mentioned are addressed.

Round 2

Reviewer 1 Report

Comments and Suggestions for Authors

Many issues have been addressed, however, many are not and therefore, there are still weaknesses.

The additions (in red) need to be checked for correct English. E.g. “how languages adapt” Adapt to what?

Paragraph 2 in the introduction cannot be well understood.

The remark on language models was not related to LLMs, but to language models as probability distributions. Check e.g. https://dl.acm.org/doi/abs/10.1145/319950.320022

This addition is not useful for the reader: In recent years, large language models (LLMs) have exploited data availability for several applications in artificial intelligence (A.I.) [31]. While our work is related and might contribute to specific problems in A.I., this exploration is beyond the scope of this paper

Better add some ideas for future research with LLMs into the conclusion and future work section.

Rank diversity is one statistical method that can be used. However, there is more discussion necessary on what it tells us and how it relates to other statistical measures. It is good that the figure on Zipf has been introduced. It shows that the order of the languages is different than in the graphs on rank diversity.

Still, rank diversity actually gives very little information about a probability distribution. More adequate and established methods would be to compare the language models (e.g. with Kullback Leibler divergence) or rank correlation coefficients. Include this or show in a discussion how rank diversity is superior to these metrics.

This is also not convincingly addressed. There needs to be work on the statistical patterns of these languages: Check previous publications on differences between Spanish and English variants and show how your work relates to them regarding method and results. 

Check the Abbreviations.           

Scale still appears in Figure 8

Comments on the Quality of English Language

some copy editing is required, especially in the newly added parts

Reviewer 2 Report

Comments and Suggestions for Authors

The authors have considered my previous comments, and updated their paper.

The authors have added some new material, including a plot of Zipf ranking exponents, to the introduction. This doesn't detract at all, and adds an interesting example of other ranking phenomena.

In my previous reading of the paper, I missed the paragraph about maintaining equal sample sizes, in order to mitigate this confounding effect. So I withdraw my previous criticism on this point. Thanks to the authors for pointing this out.

There are still a couple of points that I am not satisfied about, but they are not major threats to the validity of the work:

1. Regarding the results for Canada vs "Spanish-speaking countries", perhaps the authors have misunderstood my question. The added paragraph, while perfectly reasonable, doesn't answer my point.

I will try to present it in a different way. Regarding Figure 5, the authors state:

"Specifically, the curves for Spanish-speaking countries exhibit close
alignment, forming a cohesive cluster, while Canada displays noticeable deviation from this pattern."

I don't understand why they specifically focus on Spanish-speaking countries as exhibiting close alignment: *all* the countries except Canada exhibit similarly close alignment. As such, I don't see any distinction at all between Spanish speaking countries and countries where other languages are spoken. This spurious distinction between Spanish and English is repeated in the conclusions. This stretched interpretation of the results should be cleared up.

2. The updated paragraph on page 2 begins with a discussion about language change and evolution:

"Analyzing language change over time intervals enhances our understanding of language evolution and its influencing factors."

which I think implies a misleading connection between the current work and language change and language evolution. There is nothing in this work that specifically addresses language change.

"This method is valuable for examining geographical, grammatical, and temporal language variants, revealing how languages adapt and diverge
across different regions and periods."

This continuation exacerbates the problem: this study does not study language change over time intervals. It studies statistics of language use as a function of very short time intervals. Language change does not occur on this time scale.

At best, this kind of study might reveal differences between regions which might be a clue to historical changes, but these are not addressed even a little bit in this paper.

What the authors are doing is applying their rank diversity analysis to a new dataset, which is fine in itself, and what the authors indeed state as being their aim. The comments about language change should perhaps be removed.

In summary, I am no longer concerned about the methodology or results, but I still have some concerns about how the results are interpreted and presented. These things should be clarified.

Round 3

Reviewer 1 Report

Comments and Suggestions for Authors

The concerns were addressed.

Other more dep concerns are related to the method, but this can be subject to a scientific debate. I do not feel that this is a reason for rejection

Comments on the Quality of English Language

is Ok

Reviewer 2 Report

Comments and Suggestions for Authors

I am not really satisfied with the responses to my remaining 2 points, but I don't want to drag you into a never-ending cycle of re-revision, so I will try to find some common ground.

Regarding point 1:

Now I think I understand a bit better what you are trying to say. At first I thought you meant that the Spanish-speaking countries formed a cluster *separated* from the other countries. Instead I realised that you are saying that the Spanish-speaking countries are more tightly clustered.

This now makes your previous comments about second languages make more sense.

I appreciate your efforts in providing Table 1, but I am not convinced that the result is as clear cut as you claim. The differences are indeed slightly larger on average between the non-spanish speaking countries (I would hesitate to call any of them english speaking except perhaps  the UK, as, as you rightly point out, english is one of numerous languages present in these countries), but one could easily lump India together with Argentina, Mexico, Spain and Colombia, so what would be the differentiating factor then?

I would suggest reverting closer to the previous version here, and modifying the description along the lines of

"The spanish-speaking countries appear to be more tightly clustered in general than the other countries studied", and then after noting (as you presently do) that the explanation is not clear, one might speculate on the presence of other languages in these countries.

Regarding point 2:

Again, I'm still not convinced at all by the slightly modified paragraph on page 2. But again, trying to find a resolution, I think we might be facing a difference with regard to terminology. "Language change" is a well defined concept, meaning changes in language that occur over years, decades and centuries, and which accumulate to produce language evolution. What the authors are studying are *differences* between language use at different scales at the same point in time. This is not language change, and not related to language evolution directly, although I suppose one can't rule out that it might turn out to give a hint to some mechanism that might contribute to language evolution, although no such connection is even hinted at in this paper.

I suggest that you write "differences" instead of "change" on line 26 and following, and remove the suggestion of a connection to language evolution, which has not been established.

If you can accept these suggestions, or something similar, I'll be happy to recommend the paper for publication.
